# Digital Learning Environments in Higher Education: A Literature Review of the Role of Individual vs. Social Settings for Measuring Learning Outcomes

**Elke Kümmel [1],\* , Johannes Moskaliuk [1], Ulrike Cress [1,2] and Joachim Kimmerle [1,2]**

[1]   Knowledge Construction Lab, Leibniz-Institut für Wissensmedien, Schleichstraße 6, D-72076 Tübingen, Germany; j.moskaliuk@iwm-tuebingen.de (J.M.); u.cress@iwm-tuebingen.de (U.C.); j.kimmerle@iwm-tuebingen.de (J.K.)

[2]   Department of Psychology, Eberhard Karls University, Schleichstraße 4, D-72076 Tübingen, Germany

\*   Correspondence: e.kuemmel@iwm-tuebingen.de

**Abstract:** Research on digital learning environments has traditionally applied either an individual perspective or a social perspective to learning. Based on a literature review, we examined to what extent individual or social perspectives determined the learning outcome variables that researchers have used as measurements in existing studies. We analyzed prototypical approaches to operationalize learning settings (individual vs. social) published in peer-reviewed journals and identified their relation to several measures of learning outcomes. We rated $n = 356$ articles and included $n = 246$ articles in the final analysis. A total of 159 studies (64.6%) used an individual learning setting, while 87 studies (35.4%) used a social learning setting. As learning outcome measures, we observed self-reports, observable behavior, learning skills, elaboration, personal initiatives, digital activity, and social interactions. The two types of learning settings differed regarding the measurement of elaboration and social interactions. We discuss of the implications of our findings for future research and conclude that researchers should investigate further measures of learning outcomes in digital learning settings.

**Keywords:** digital learning environments; higher education; evaluation methodologies; learning outcomes; media in education; collaborative learning

## 1. Introduction

The efficient use of digital learning environments in higher education is an important research topic from both a scientific and a practical perspective. Learning in digital learning environments is characterized by the provision of learning materials that are independent of time and location, and by broad access to learning materials. Moreover, digital learning environments also support educational opportunities for all types of learners and provide digitally-enhanced instruction [1–3]. Educational researchers from diverse disciplines have been trying to identify the success factors of learning with digital media in higher education for about two decades [4–10]. One central aim of higher education is to foster students' potential for high-quality accomplishments [11–14] and support them in applying their knowledge to future challenges in their professional lives [15,16]. Therefore, research on the use of digital learning environments in higher education should pay particular attention to learning outcomes as a prerequisite for evaluating learning success.

There are two main reasons why researchers and practitioners recommend the use of digital learning environments in higher education. First, in an increasingly digitalized world, education needs to be digital as well [17–20]. Students should be encouraged and empowered to use digital media for

communication and collaboration as well as for learning and knowledge exchange in an appropriate way to become competent and proficient members of a knowledge society. Second, digital learning environments promise to make learning and teaching more effective, for example, by increasing learners' motivation [14,21], adapting to students' prior knowledge [16], or providing the possibility for mobile and ubiquitous learning [22,23].

However, the findings of existing studies on the impact of digital media on learning are ambiguous [24–27]. In general, influencing factors, such as teachers [28,29], prior knowledge [30,31], or the novelty of the particular digital setting [32] seem to have greater effects on learning outcomes than the use of digital media per se. One reason for marginal findings on the effects of digital media in these studies might be that they are highly heterogeneous with regard to measurements and the learning settings that they applied. Therefore, the study presented here summarizes common measurements of variables that capture learning outcomes in existing empirical studies. This contributes to finding a common language of researchers to describe effects by having a shared understanding of distinctive learning outcomes. We also argue that the particular theoretical perspective that researchers and practitioners take toward learning with digital media may have an impact on how they design learning environments, how they operationalize relevant variables, and how they measure learning outcomes [26]. Research on digital learning environments has traditionally applied two perspectives of examining and understanding how people learn [33]: A cognitive, individual-oriented perspective that focuses on individual cognition, and a social, community-oriented perspective that focuses on distributed cognition and collaboration [34–36]. The cognitive perspective has been upheld mainly in psychology and in cognitive science research, while the social perspective has been the dominant approach in the learning sciences for roughly 30 years now [37].

The objective of the study presented here was to examine how a cognitive perspective compared to a social perspective determined the dependent variables that researchers have used in existing studies. The goal of this approach is to comprehend the role that these theoretical perspectives play in the design of digital learning environments and the evaluation of learning outcomes.

## 2. Theoretical Perspectives on Learning and Research Question

Understanding the importance of these theoretical perspectives is one precondition for transferring scientific results into educational practice. In the following sections, we summarize the key ideas, concepts, and methods of the individual and social perspectives on digital learning environments and introduce our study idea.

### 2.1. Individual Perspectives on Learning

Individual perspectives deal with individual information processing and focus on individual thinking, including attention, mental representation, learning, memory, problem solving, and decision-making [38,39]. From this standpoint, learning can be described as selecting information and acquiring knowledge through the encoding, storage, and retrieval of information. Learning activities of a single learner would then involve, for example, content-specific examination of learning materials (e.g., leading to understanding), achieving a certain knowledge state (e.g., leading to a test result), or individually creating a previously defined product (e.g., leading to an essay or a work object). From this perspective, digital media can be used to adaptively provide learning content and instructions. Beyond studying how students learn, it is relevant to understand how learners can be instructed or supported [40–42].

Cognitive theorizing also takes meta-cognitive monitoring into account, as this is a major aspect of self-regulated learning [43]. Cognitive models often rely on training, problem solving, or computational thinking [44–46]. They emphasize strategies for instructing learners to understand new information, construct mental representations of knowledge, and integrate information into cognitive schemas [47]. Cognitive processes are mechanisms that induce learning depending on the mental capacity of

learners [48–50]. With the development of digital media, learning environments can be designed in such a way that they can rise to the challenge of meeting a learner's current cognitive load.

To predict learning from a cognitive perspective, researchers have investigated the process of knowledge acquisition where learners create mental representations of their knowledge [51–54]. From a cognitive perspective, a learner's memory and cognitive capacity [55–58], attention [59], or decision-making [60] are characteristics which determine learning. Research on individual learning with digital media, for example, indicated that dealing with digital learning material implies handling multimodality and interactivity [61,62], and that digital learning material is associated with specific e-tools [63] or virtual elements [32,64].

### 2.2. Social Perspectives on Learning

The social perspective postulates that learning is strongly influenced by the social environment in which it occurs [65]. This assumption is in line with the social constructivist theory developed by Vygotsky [65] and indicates that learners need to be actively engaged in their social environment. As a consequence, learning can be conceptualized as a cooperative or collaborative endeavor [66]. From this perspective, individual learning is socially mediated and not independent of the social context it is embedded in. On the contrary, the individual learners' cognitive systems strongly interact with social systems [67]. Learning (as an individual process) and knowledge construction (as a collective process) depend on knowledge-related activities that arise through socio-cognitive conflicts between these two systems [36]. Thus, communication and social interaction may trigger learning and knowledge construction [68–70].

To solve a task for the first time, learners need scaffolding and support from peers or teachers [71–73]. If two or more people work together in computer-supported collaborative learning (CSCL), CSCL researchers tend not to focus on what happens in a single learner's cognitive system. They rather take the interaction among people or within the CSCL environment into account [74]. Learners are part of a social context and they learn how to act and interact within this situation. Such learners make important contributions in two respects. On the one hand, they internalize knowledge and develop as an individual; and on the other hand, they react to other people and externalize their knowledge into the social context [36]. From a social perspective, learning activity and learning outcomes are strongly dependent on the interaction within the group.

Research on social learning in digital learning environments indicates that dealing with digital media implies the development of social awareness [75]. This involves making use of social media and social communication technology [76,77], and defining one's role in a digital network [20,78]. Moreover, social awareness can be developed by using specific e-tools [79] or virtual elements [64,80] to influence learning in a social setting.

### 2.3. The Research Presented Here

To answer the question how learning outcomes have been measured in empirical studies on learning with digital media in higher education, it is relevant for educational research and practice to understand what perspectives researchers have on learning and, as a consequence, how they tend to measure learning outcomes [19]. In the study presented here, we focused on individual and social approaches in existing empirical research in order to identify whether there is a relationship between the learning setting that researchers have chosen for their studies in higher education and the particular variables they have measured. Educational researchers tend to operationalize a learning setting in their research according to their own general perspective on learning, as described above. For this reason, we identified the learning setting of each study and considered which learning outcomes were measured. This approach provided an overview of existing empirical research, which can serve as a basis for further discussion. In identifying certain gaps in the research, it furthermore suggests where future studies need to focus.

We hypothesized that there would be a relationship between the general perspective and the particular learning outcomes in such a way to show that researchers use different dependent variables in their learning settings due to their particular perspectives. To examine potential co-occurrence of learning perspective and learning outcomes on a basis with solid data, we gathered data from published studies on digital learning environments in higher education. In the following section, we describe the method of our empirical procedure in detail.

## 3. Method

### 3.1. Searching the Literature: Article Selection

Following the procedure proposed by Cooper [81], we analyzed which learning outcomes previous research has captured in the context of digital learning environments in higher education. We used the standard Web of Science search (ISI Web of Knowledge, Clarivate Analytics). We selected the categories education, psychology, and computer science and identified four relevant topics for our search: (1) digital learning environments, (2) instructional design, (3) higher education, and (4) performance criteria. The first string ensured a neutral perspective which enabled us to find studies about digital learning environments. The second string focused on the instructional perspective, as we aimed to examine processes of learning and teaching. The third string restricted the search to higher education and academic performance [82]. Performance criteria refer to the proficiency of a learner or a group in a given task (e.g., individual or collaborative) and the resulting activities for accomplishing this task. Consequently, the fourth string aimed at identifying performance criteria that are frequently considered for evaluating achievement in higher education. The search procedure and the inclusion criteria for the articles considered for analysis are shown in Figure 1.

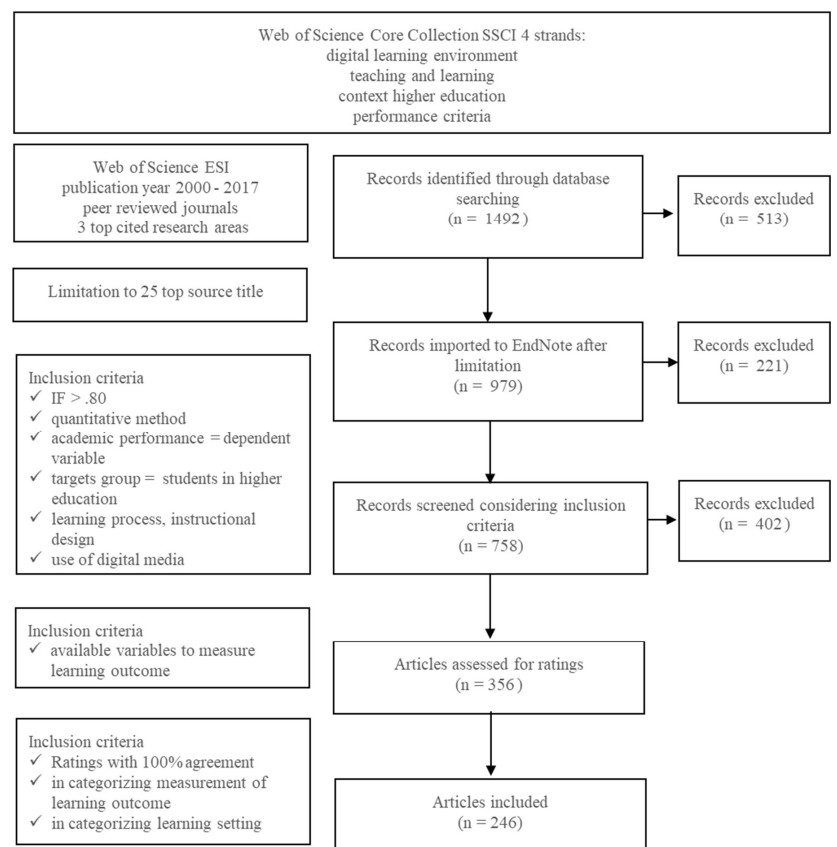

**Figure 1.** Search procedure and inclusion criteria for the articles considered for analysis.

*3.2. Gathering Information from Studies: Coding Guide*

Our variables of interest were measures of learning outcome (seven categories as described below) and learning setting (individual / social orientation). Independent raters assessed this information in all of the 356 articles and transferred it into an SPSS coding sheet (the rating procedure is described in Section 3.3). The availability of the information was substantial for an article's inclusion in further analyses. Articles without detailed information about these particular variables of interest were excluded.

### 3.2.1. Learning Setting: Individual vs. Social Orientation

In the studies we selected, we identified two different orientations in higher education learning settings that represented a researcher's perspective: An individual orientation vs. a social orientation. Studies with an individual orientation supported individual learners in digital learning environments to create a mental representation and to foster knowledge acquisition [83,84]. This orientation implied that learning activities should be affected by individual cognitive, motivational, and behavioral aspects. A study design was coded as an individual setting if its abstract revealed that learners in individual learning scenarios were assigned to an individual task that they fulfilled on their own.

In studies with a social orientation, learners' participation in social systems and the collaborative application of learning materials were key aspects of learning. This was the case, for example, if two or more people worked together in a CSCL environment [85–87]. A study design was thus coded as representing a social setting if the study abstract indicated any kind of collaborative task, either accomplished in a group or through teamwork.

### 3.2.2. Measurements of Learning Outcomes

We created categories that indicated several different options for measuring particular learning outcomes in digital learning environments. We focused on the outcomes of learning processes, knowledge construction, or knowledge-related activities. By developing meaningful categories for the measurement of learning outcomes, we met the challenge of sorting through current requirements of research in learning with digital media and acquiring first insights from the articles (i.e., records were screened with inclusion criteria in mind). We took two theoretical approaches into account that describe learning processes in digital learning environments [20,52,88,89].

Chi and Wylie [52] proposed four types of engagement activities: passive, active, constructive, and interactive. Engagement can be interpreted as a continuum of growing learning processes with predefined learning materials. The authors describe typical materials and activities which enable handling information within a digital learning environment safely and lead to success in learning. Therefore, any attempt at measuring learning outcomes should include these considerations.

Wilson et al. [90] proposed the perspective of social networking to describe learning in digital communities. They proposed a hypothetical individual learner who was embedded in a social network and who fulfilled a certain role. This goes along with the assumption that the key to learning effectiveness is to create interaction, to encourage deep reflection, and to reach definitive conclusions [91]. The authors considered different levels of performance and provided suggestions about how to order skills and competencies. We considered this also to be a potential method to measure learning with digital media.

These approaches provide theoretical frameworks to describe what is important for research in learning with digital media and offer a basis for categorizing measurements of learning outcomes. We integrated subjective (i.e., self-reports) and objective measurements (i.e., observable behavior) as well as measurements of self-regulation and knowledge changes (i.e., learning skills, elaboration). Furthermore, we integrated the current need to measure learning with digital media that emerged from the reasoning above [91], that is, measurement of interaction on a personal, technological, and context-specific level (i.e., personal initiative, digital activity, and social interaction). For each category, we first provide a

short description and theoretical assumption extracted from existing research. We then point out the role of each category for learning in digital learning environments in general and provide examples to underpin our categorizations, which we grouped into the superordinate categories of method, cognition, and activities (see Table 1).

**Table 1.** Measurements of learning outcomes in digital learning environments.

| Category | Examples | Learning Outcome is Evaluated on the Basis of … |
|---|---|---|
| Method | | |
| Self-report | Students report about their satisfaction, motivation or attitude | … experience, perception, or values of a learner. |
| Observable behavior | Enrollment or final completion of lectures or seminars | … intention, persistence or effectiveness of a learner's behavior. |
| Cognition | | |
| Learning skills | Self-regulation, awareness or writing skills | … meta-cognition. |
| Elaboration | Vocabulary-tests or transfer tasks | … cognitive measurements. |
| Activities | | |
| Personal initiative | Number of contributions to discussions or frequency of use | … mere participation or pro-activeness of a learner. |
| Digital activity | Sourcing and searching behavior | … digital maturity level or active usage of digital tools. |
| Social interaction | Collaboration with peers or communication with professors | … social influence on activities of a learner. |

1. Self-report

Self-reports reveal what individual learners think about their abilities, the learning material, the digital learning environment, or the learning outcomes they wanted to achieve [14,21,92]. Such data may be relevant for understanding the subjective side of learning, but they also carry the risk of containing biased information due to a fragile and subjective measurement.

Examples: Learners' perceptions of their own attitude, satisfaction, or motivation; self-reported information may comprise personal relevance, commitment, self-efficacy, and perceived importance or beliefs.

2. Observable behavior

Observable behavior represents the objective behavior of a learner and evaluates learning outcomes in an action-oriented manner. This measurement focuses on the goals of learners and their intention to learn. It includes activities such as choice of lectures, persistence, or efficacy to complete a course [93,94].

Examples: Passing or not passing a course, course selection, choosing a field of study or a subject of specialization.

3. Learning skills

This category relies on models of self-regulation, for example, the model of self-regulated learning that comprises three-layer levels of regulation processes [95]: regulation of the self, of the learning process, and of processing modes. At another level, according to Krathwohl [96], learning outcomes in this category refer to metacognitive knowledge, or knowledge about own cognitive processes, reflection, or self-regulation.

Examples: Reading and listening skills, awareness of group processes, or writing involved in acquiring learning skills.

4. Elaboration

Elaboration refers to learners' levels of cognitive processing. Either the surface approach, deep approach, or achieving approach were used in the learning processes [97,98]. Learning outcomes reflect these levels of processing via recall, comprehension, or application tasks [99–101].

Examples: Multiple choice tests, vocabulary tests, comprehension tests, or essays.

5. Personal initiative

Being pro-active is a core characteristic for initiating interaction with digital media. Personal initiative is helpful to begin learning in digital environments and to take advantage of educational opportunities in learning with digital media. A minimum of personal initiative is important to initiate learning [102,103].

Examples: Participation, attendance, access to email accounts, data about log-ins, quantity of sent emails, or number of contributions.

6. Digital activity

This category includes engagement in a digital learning environment and active contributions [104]. Digital activity can be measured as performance via log-files. This may include learners' accessing information, their management, integration, and evaluation of information, their search inquiries, and their use of blogs or wikis.

Examples: Help-seeking behavior, network behavior, search behavior, or reflective and conscious usage of digital environments.

7. Social interaction

Since most digital learning environments integrate several instructional designs, and even individual learning settings include communication tools and situations, this is not a category that would be limited to a social learning setting. This indicator allows considering studies that measure a kind of social interaction as a dependent variable. It includes activities that refer to any kind of cooperation or collaboration [105].

Examples: A dyad's discussion, a presentation, or a discussion outcome on an individual and a group level.

*3.3. Evaluating the Quality of Studies: Rating Procedure*

Three raters (a research associate and two student assistants) were trained to identify available variables of interest within the articles' abstracts. They received written descriptions of the categories (Table 1), written descriptions of the learning settings (see Section 3.2.1), and verbal instructions about the categorization tasks and about the coding sheet. Raters were trained to identify available dependent variables (i.e., measurements of learning outcomes) and to assign these variables to one of the seven categories of the coding guide. In those cases, in which the raters were unable to make a decision based on the abstract, they were instructed to examine the relevant information in main text. In order to ensure a high level of inter-rater agreement, we only included ratings with a full agreement among the raters.

The raters rated study design (0 = exclude article, 1 = individual, 2 = social) and measures of learning outcomes (0 = not available, 1 = self-report, 2 = observable behavior, 3 = learning skills, 4 = elaboration, 5 = personal initiative, 6 = digital activity, 7 = social interaction). An article's abstract could reveal more than one measurement of learning outcomes; in this case, several codes were assigned.

The rating procedure needed to distinguish two methodological approaches: on the one hand, we identified available unambiguous descriptions of one learning setting per article (i.e., individual or social). On the other hand, we identified dependent variables within these articles (i.e., measure of learning outcomes). We created word clouds to visualize word frequency using the website https://tagcrowd.com. We excluded filling words and used a maximum of 75 words in the word clouds.

## 4. Results

We first present the findings regarding the learning settings. Then we provide the results of the dependent variables used in the articles, that is, the measurements of learning outcomes. Finally, we present the findings regarding our general research question.

### 4.1. Learning Settings

As described above, we identified *n* = 246 articles with a clear description of a learning setting (see Figure 1). The digital learning setting of 159 articles revealed an individual orientation and 87 articles revealed a social orientation in their study design. Descriptions of the individual and social learning settings can be seen in Figure 2. These depictions show that studies with a social learning setting used more varied terms to describe variables. While more studies used an individual learning setting, the word cloud of the social settings include more as well as more diverse words.

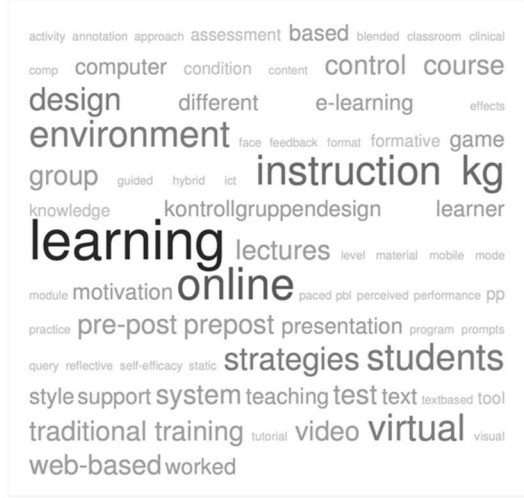
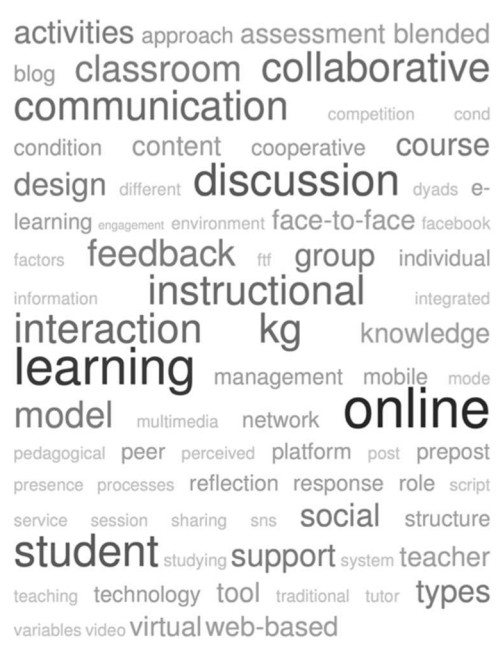

**Figure 2.** Word clouds for individual (left) and social (right) learning settings.

The *n* = 246 articles were published in 14 different peer-reviewed journals. The journals Computers & Education (76), BMC Medical Education (31), and Educational Technology & Society (27) provided most of the identified articles. Studies from journals such as Computers & Education and BMC Medical Education used more individual than social learning settings, whereas studies in The Internet and Higher Education and the Australasian Journal of Educational Technology described more social than individual settings (Table 2).

**Table 2.** Learning settings (individual vs. social) represented in the journals considered for the analysis.

| Journal | Learning Setting | | Total |
|---|---|---|---|
| | Individual | Social | |
| Advances in Health Sciences Education | 2 | 3 | 5 |
| Assessment & Evaluation in Higher Education | 0 | 3 | 3 |
| Australasian Journal of Educational Technology | 6 | 12 | 18 |
| BMC Medical Education | 35 | 9 | 44 |
| British Journal of Educational Technology | 13 | 11 | 24 |
| Computers & Education | 79 | 23 | 102 |
| Educational Technology & Society | 16 | 12 | 28 |
| Educational Technology Research and Development | 8 | 0 | 8 |
| Instructional Science | 5 | 1 | 6 |
| Interactive Learning Environments | 11 | 6 | 17 |
| International Review of Research in Open and Distance Learning | 6 | 2 | 8 |
| Internet and Higher Education | 8 | 12 | 20 |
| Journal of Computer Assisted Learning | 8 | 7 | 15 |
| Journal of Science Education and Technology | 7 | 1 | 8 |
| | 204 | 102 | 306 |

## 4.2. Measurements of Learning Outcomes

In total, raters identified $n = 306$ dependent variables for a measurement of learning outcomes from the $n = 246$ articles. Self-report (128) and elaboration (113) were captured most frequently as dependent variables in individual as well as social learning settings, while the least used measurements were personal initiative (1) and observable behavior (9) (Table 3).

**Table 3.** Absolute frequency of measures of learning outcomes (n = 306) in individual and social learning settings.

| Learning Setting | Learning Outcomes | | | | | | | |
|---|---|---|---|---|---|---|---|---|
| | 1 | 2 | 3 | 4 | 5 | 6 | 7 | |
| individual | 79 | 7 | 17 | 86 | 0 | 14 | 1 | 204 |
| social | 49 | 2 | 9 | 27 | 1 | 4 | 10 | 102 |
| | 128 | 9 | 26 | 113 | 1 | 18 | 11 | 306 |

*Note.* Learning setting (individual vs. social) and measurement of learning outcomes (1 = self-reports, 2 = observable behavior, 3 = learning skills, 4 = elaboration, 5 = personal initiative, 6 = digital activity, 7 = social interaction).

We provide examples of measures of learning outcomes from the identified articles. We also provide word clouds for the two most frequently used dependent variables self-report (Figure 3) and elaboration (Figure 4). The word frequencies of dependent variables that were used to measure learning outcomes are depicted in these visualizations.

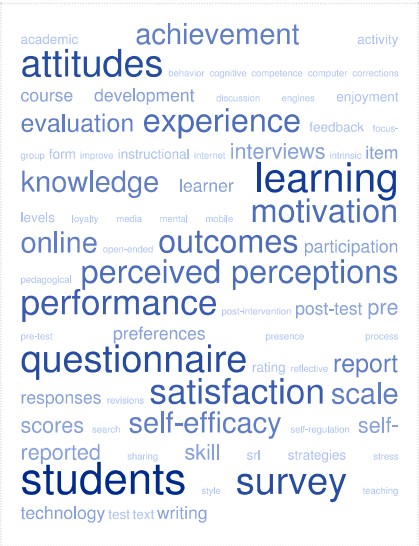

**Figure 3.** Word cloud for the dependent variables in the category self-report.

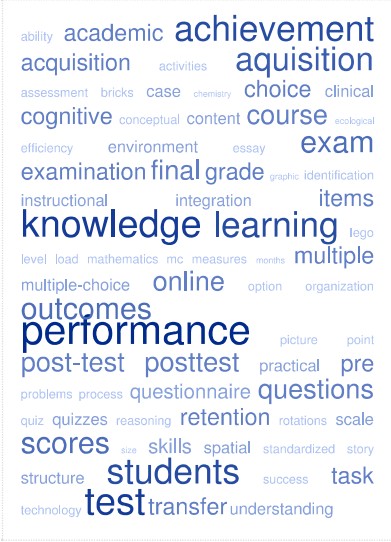

**Figure 4.** Word cloud for the dependent variables in the category elaboration.

An example of a study using self-report to measure learning outcomes reported self-efficacy beliefs and intrinsic motivation [32]. Studies that measured learning outcomes through observable behavior used variables such as drop-out rates [106], number and duration of sessions [107], completion of exams [108], or class attendance [109]. Examples of measures in the category learning skills were intercultural communicative competence, intercultural awareness and intercultural knowledge [110], or learners' reflection levels [111]. Examples of elaboration measures were exams on lecture content [112] or problem-solving activities [113]. We identified one measure of learning outcome that represented personal initiative on a social level: Shea et al. [114] examined learning presence through social network analysis and quantitative content analysis in a student public class discussion (i.e., personal initiative), as well as private products of knowledge construction (i.e., elaboration). For digital activity, we identified tracking systems [115] and search activities [116] as measures of learning outcomes. Studies rated within the category social interaction used measures such as team-learning outcomes [117], mutual feedback [118], or team discussions [119].

### 4.3. Measures of Learning Outcomes in Individual and Social Learning Settings

Across all categories, a chi-squared test showed that measures of learning outcomes differed significantly between the chosen learning settings, $\chi 2$ (6, 306) = 25.89, $p < 0.001$ (Tables 3 and 4). In social learning settings researchers used elaboration significantly less frequently than in individual learning settings, binom (27, 113, *prob* = 1/3), $p = 0.036$. Elaboration was the favorite measure of learning outcomes for researchers with an individual research approach. We also found that the category with studies that measured social interaction as a dependent variable was chosen for social as well as for individual learning settings, but that these social interaction measures were more frequently used in social than in individual learning settings, binom (1, 11, *prob* = 1/3), $p < 0.001$. There were no significant differences for the other categories (Figure 5).

**Table 4.** Relative frequency of measures of learning outcomes (n = 306) in relation to learning setting in percent.

| Learning Setting | Learning Outcomes | | | | | | |
|---|---|---|---|---|---|---|---|
| | 1 | 2 | 3 | 4 | 5 | 6 | 7 |
| individual | 38.73 | 3.43 | 8.33 | 42.16 | 0 | 6.86 | 0.49 |
| social | 48.04 | 1.96 | 8.82 | 26.47 | 0.98 | 3.92 | 9.80 |

*Note.* Learning setting (individual vs. social) and measures of learning outcomes (1 = self-report, 2 = observable behavior, 3 = learning skills, 4 = elaboration, 5 = personal initiative, 6 = digital activity, 7 = social interaction).

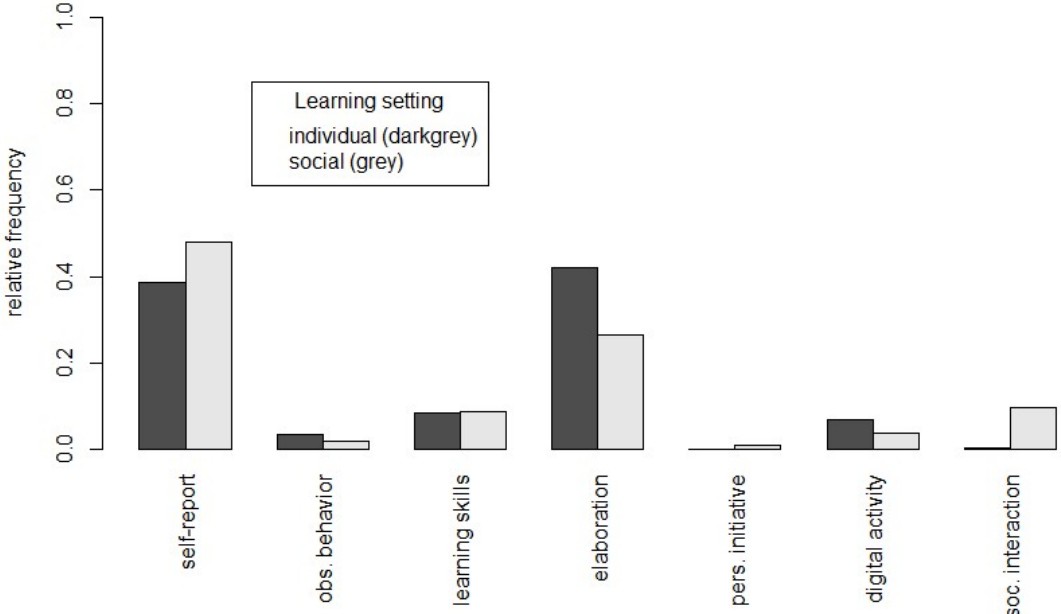

**Figure 5.** Relative frequency of measures of learning outcomes in individual and social learning settings.

## 5. Discussion

In the study presented here, we focused on cognitive and social approaches in existing empirical research in order to identify whether there was a relationship between the learning settings and the measurement of learning outcomes that were applied by researchers. In each study, we identified the general design of digital learning environment and, at the same time, considered the respective measures of learning outcomes that were used in that study. In total, there were more studies from researchers who used individual settings than studies with social settings. We also found that self-reports and elaboration were captured most often as measures of learning outcomes. We had hypothesized that there would be a relationship between the general perspective (individual vs. social learning setting) and the particular measurement of learning outcomes, which would indicate that

researchers use different dependent variables depending upon their particular research approach. As hypothesized, we found that the measures of learning outcomes in studies with an individual learning setting differed from the measures in studies with a social learning setting. Researchers with an individual approach used different variables to evaluate learning outcomes compared to researchers who used a social setting. The comparison of studies with individual and social settings revealed that the measure elaboration was used relatively more often in studies with individual settings than in studies with social settings. The measure social interaction, in contrast, was used less often in studies with individual settings than in studies with a social approach.

*5.1. Heterogeneous Evaluation of Learning Outcomes*

This literature review has generated a broad variety of performance criteria as indicators for measuring learning outcomes in higher education. The results regarding learning outcomes with digital media in higher education showed that the classification of learning outcomes is not consistent in previous studies. Even for the selected and narrow context of higher education, the terminology for learning outcomes is heterogeneous. However, the database search produced many results of high-quality studies. There seem to be overall favorite variables for measuring learning outcomes, like elaboration and self-reports.

*5.2. Learning Outcomes in Learning Settings*

Researchers with an individual perspective used elaboration relatively more often to evaluate learning than researchers with a social orientation. This is comprehensible, as learning activities in an individual learning setting often explicitly refer to recall, comprehension, or cognitive performance for predicting achievement or success. In addition, individual processes that are particularly relevant from an educational viewpoint are learning and memory processes in terms of selecting information and acquiring knowledge through encoding, storage, and retrieval of information. So individual learning is usually seen from a cognitive perspective [120]. Our analysis disclosed a relative frequency of 42.16% to measure elaboration in studies with an individual learning setting. Nevertheless, the analysis also found the usage of variables to describe elaboration in studies that represented a social perspective with a relative frequency of 26.47%. Overall, 113 out of 306 ratings (36.93%) indicated measures of elaboration. Elaboration appears to be a measure that is highly relevant for both individual and social approaches.

Self-report was also a very frequently used category. Altogether, in 41.83% of the studies, raters identified self-reports as outcome measures. Self-reports seem to be highly relevant for understanding what learners think about their abilities, the learning material, the digital learning environment, or the learning outcomes they wanted to achieve irrespective of whether researchers used an individual or social setting. In total, 241 out of 306 measurements belonged to the self-report or elaboration category. These two measurements do not seem to be particularly tailored to digital settings, however. Self-reports and elaboration are relatively traditional measures and do not really take digital characteristics into account.

We had expected to find more measures of learning outcomes in the activity category, which included personal initiative and digital activities as these have a specifically digital focus. These activities are supposed to be highly relevant in the digital world, but only 30 out of 306 measurements belonged to these categories. Network-specific access to learning institutions in higher education, digital learning material, and digital communication tools as well as digital learning environments have become more and more prevalent, and the need for research in this context has grown. Therefore, the low frequency of variables that measured digital activity in both individual and social learning settings was surprising. We hope that future studies will take these variables into account more frequently.

## 5.3. Limitations

Our findings may have been affected by the selection of the journals in our database search. We only included peer-reviewed studies, and peer-reviewed journals have specific aims and scopes. For example, BMC Medical Education focuses on training and evaluation of performance, such as grades, which may promote research about learning progress of individual learners. This focus on the individual level and the cognitive performance of learners could explain why there were more studies from an individual than from a social perspective (Table 2). At the same time, however, this is a pity, because social settings are of course also very important for medical training [121]. This applies, for example, to the field of doctor-patient communication [122,123] or to inter-professional cooperation [124]. Computers & Education is additionally interested in field studies with interventions in designs with pre- and post-tests of individual learners or control groups to compare digital vs. non-digital learning environments. Testing and comparing these pedagogical issues of digital technology as they pertain to individuals also could lead to publishing more studies of researchers with an individual than a social orientation. However, the focus on interventions should not prevent researchers from studying the role of social variables and settings in the future.

Furthermore, the gathered data resulted from identifying particular learning outcomes and did not take other potentially relevant aspects into account. For example, the data did not consider the discussions of the articles. Therefore, it could be a sensible next step to carry out more comprehensive analyses of empirical studies that would provide detailed descriptions of context, design, dependent and independent variables, statistics, and effects of the performance criteria in individual and social learning settings. Moreover, it could be promising to collect data about independent variables that tend to co-occur. Future studies could also analyze the structure of the studies and conduct statistical investigations to predict successful outcomes with digital media in higher education.

## 5.4. Balancing Perspectives

We argue for balancing research approaches that deal with learners' development on an individual level as well as the development of the social context in higher education. So far, many educational researchers have tended to examine either an individual or a social learning setting in their research. We hope that our results can be used in future research that aims to bring both approaches together. Furthermore, the interrelatedness between individual and social processes makes it sometimes difficult to distinguish among various knowledge-related activities, and researchers need to disentangle influencing effects and variables of interest. We postulate that this is an iterative process where researchers should reflect upon their own perspective in a responsible manner and make it transparent to others.

With respect to interdisciplinary research and collaboration projects, it could be promising to define commonalities between different research approaches. According to O'Connor and Allen [15], learning should take into account that learners aim to form an identity and to become community members. The authors emphasize that learning must be considered to be a relational phenomenon. As a result, it could be a goal of higher education to train students for a role within a professional network. From this point of view, it seems to be the social system and the individual learner who would benefit from an adequate fit of an educational system, a digital learning environment, and measurements of learning outcomes [20,29,90]. This could create a closer link between research and teaching, support the ongoing change process for institutional implementations of digital learning environments, and be a contribution to the ongoing challenge of adequately preparing teachers for digital teaching environments.

## 6. Conclusions

We provided an overview of empirical studies of digital environments in higher education and the learning outcomes they have measured. As hypothesized, results revealed overall that there was a relationship between the learning setting that was applied (individual vs. social orientation)

and the variables used to measure learning outcomes. We found support for our categorizations of measuring learning outcomes in terms of self-reports, observable behavior, learning skills, elaboration, personal initiative, digital activity, and social interaction. The analysis revealed two particularly popular measures of learning outcomes (i.e., self-reports and elaboration).

Our approach of gathering and structuring a huge amount of data from high-quality empirical studies may enable practitioners and scientists to rely on and refer to the results. We identified variables and categorized measures of learning outcomes of students. The results provide first insights into frequently used measures. This study was a first step in the direction of investigating this research topic. In sum, the goal of describing measures of learning outcomes was achieved, and the chosen categorization to describe evaluations of learning outcomes of individuals provide a foundation for further study. With respect to the properties of digital learning environments, future studies should try to elaborate on and potentially revise these categorizations.

For future research, we recommend more creative measurements of variables to evaluate learning outcomes in digital learning settings in the context of higher education. For this purpose, researchers need to carefully reflect upon their research subjects and study designs in digital learning environments and think about how to deal with measuring learning. Disentangling influencing effects and independent variables would be helpful to make interdisciplinary educational research sustainable for the future.

**Author Contributions:** Conceptualization, E.K. and J.K.; methodology, E.K.; software, E.K.; validation, E.K. and J.M. and U.C. and J.K.; formal analysis, E.K.; investigation, E.K.; data curation, E.K.; writing—original draft preparation, E.K.; writing—review and editing, E.K. and J.K.; visualization, E.K.; supervision, J.K. and U.C.; project administration, J.M.; funding acquisition, U.C. and J.M.. All authors have read and agreed to the published version of the manuscript.

**Funding:** This research was funded by Bundesministerium für Bildung und Forschung: 16DHL1010.

**Conflicts of Interest:** The authors declare no conflict of interest. The funders had no role in the design of the study; in the collection, analyses, or interpretation of data; in the writing of the manuscript, or in the decision to publish the results.

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
