# Peer review of "Digital Learning Environments in Higher Education: A Literature Review of the Role of Individual vs. Social Settings for Measuring Learning Outcomes"

_education, doi:10.3390/educsci10030078_

Round 1
Reviewer 1 Report
ABSTRACT
In the Abstract it would be helpful to state that 356 studies were reviewed, as well as the proportions of individual/social.
THEORETICAL PERSPECTIVES
The categorisation of literature into individual and social perspectives is well supported. The former derives from cognitivism as a theory of learning, and the latter is surely social constructivist –– although this term is nowhere to be found in the overview of socially-mediated learning in §2.2.
METHOD
The four-string procedure for article selection and the criteria for journal selection are appropriate and systematic.
There are no details of how the three raters mentioned in §3.3 were trained, and whether procedures were implemented to optimise inter-rater reliability prior to data collection. More information should be provided here.
DISCUSSION
The outcomes of this study could be more revealing of the epistemological and ontological standpoints of the authors of the respective target articles. Do such authors always 'stay' in a particular individual/social perspective across successive articles? Table B2 is interesting, with substantial differences in learning setting for BMC Medical Education and Computers & Education: what does this say about the discipline and backgrounds of the authors that choose these journals and the journals that choose these authors?
OVERALL
This is a worthwhile and interesting paper that makes a significant contribution to literature in the area and provides helpful suggestions for the refinement of future literature reviews.
Reviewer 2 Report
The main objective of the author(s) is to examine to what extent individual or social perspectives determined the learning outcome variables. Specifically, they focused on individual and social approaches in existing empirical research in order to identify whether there is a relationship between the learning setting that researchers have chosen for their studies in higher education and the particular variables they have measured.
First of all, why do you have a related work section? I understand that you made a small discussion in your intro but there exist various works in the literature that deal with individual and/or social perspectives in digital learning environments. These works might not all related to higher education but you can present their similarities or differences with your own work.
You are writing huge sentences that are difficult for the readers to read. Also, the meaning of many sentences is confusing.
For example, see the next one:
- “Learning in digital learning environments is characterized by the provision of learning materials which are independent of time and space, broad access to learning materials, support of educational opportunities also for non-traditional learners and by digitally-enhanced instruction”.
- The next sentence need to be rephrased. “Therefore, the study presented here summarizes common operationalizations of variables measuring learning outcomes in existing empirical studies”.
- Also, you are using many times the word “operationalizations” in your manuscript. I do not know if this is a suitable word.
- “To answer the question how learning outcomes have been measured in empirical studies on learning with digital media in higher education, it is relevant for educational research and practice to understand what perspectives researchers have on learning and, as a consequence, how they tend to measure learning outcomes”
- “This category represents being alert in a digital learning environment and interacting responsibly” – Confusing meaning!
There are many confusing sentences in the manuscript. Please read carefully the manuscript and correct all these sentences.
In general, the meaning that comes out of the Introduction section is quite confusing. Please check it.
Also, please check your manuscript for grammar and syntax errors. For example, you are writing “…own general perspective on learning, as decribed above…..” – You mean “described”.
Further, simple writing is not good enough all the time. For instance, you are writing: “If two or more people work together in computer-supported collaborative learning (CSCL), it is not so relevant what happens in the head of a single learner.” – “happens in the head….”? Too simple!!
I think that you do not need to be so detailed on how you have selected the articles for your research. You may present briefly the section 3.1. Also, the wordclouds A2, A3, and A4 do not add value to your article.
Round 2
Reviewer 2 Report
The author(s) addressed all my comments.